# Effect of Glutamine Supplementation on Muscular Damage Biomarkers in Professional Basketball Players

**DOI:** 10.3390/nu13062073

**Published:** 2021-06-17

**Authors:** Alfredo Córdova-Martínez, Alberto Caballero-García, Hugo J Bello, Daniel Pérez-Valdecantos, Enrique Roche

**Affiliations:** 1Departamento Bioquímica, Biología Molecular y Fisiología, Facultad de Ciencias de la Salud, GIR: “Ejercicio Físico y Envejecimiento” Universidad Valladolid, Campus Universitario “Los Pajaritos”, 42004 Soria, Spain; danielperezvaldecantos@gmail.com; 2Departamento de Anatomía y Radiología, Facultad de Ciencias de la Salud, GIR: “Ejercicio Físico y Envejecimiento” Universidad Valladolid, Campus Universitario “Los Pajaritos”, 42004 Soria, Spain; albcab@ah.uva.es; 3Departamento Matemáticas, Escuela de Ingeniería de la Industria Forestal, Agronómica y de la Bioenergía, GIR: “Ejercicio Físico y Envejecimiento” Universidad Valladolid, Campus Universitario “Los Pajaritos”, 42004 Soria, Spain; hjbello.wk@gmail.com; 4Instituto de Bioingeniería y Departamento de Biología Aplicada-Nutrición, Universidad Miguel Hernández, 03202 Elche, Spain; eroche@umh.es; 5Instituto de Investigación Sanitaria y Biomédica de Alicante (ISABIAL), 03010 Alicante, Spain; 6CIBER Fisiopatología de la Obesidad y Nutrición (CIBEROBN), Instituto de Salud Carlos III (ISCIII), 28029 Madrid, Spain

**Keywords:** exercise, glutamine, muscle damage, recovery, supplementation

## Abstract

Scientific evidence supports the role of L-glutamine in improving immune function. This could suggest a possible role of L-glutamine in recovery after intense exercise. To this end, the present report aimed to study if oral L-glutamine supplementation could attenuate muscle damage in a group of players of a mainly eccentric sport discipline such as basketball. Participants (*n* = 12) were supplemented with 6 g/day of glutamine (G group) or placebo (P group) for 40 days in a crossover study design (20 days with glutamine + 20 days with placebo and vice versa). Blood samples were obtained at the beginning and at the end of each period and markers from exercise-induced muscle damage were determined. The glutamine supplemented group displayed significantly low values of aspartate transaminase, creatine kinase and myoglobin in blood, suggesting less muscle damage compared to the placebo. In addition, adrenocorticotropic hormone levels were lower in the glutamine supplemented group than in the placebo. As a result, the circulating cortisol levels did not increase at the end of the study in the glutamine supplemented group. Altogether, the results indicate that glutamine could help attenuate exercise-induced muscle damage in sport disciplines with predominantly eccentric actions.

## 1. Introduction

During intense exercise, the muscles develop fatigue and weakness throughout a limited period of time. The process is exacerbated in the case of subjects that do not exercise regularly, as well as during very intense exercise bouts. In these particular cases, muscle damage may take days to recover. This type of muscle damage is commonly known as “exercise-induced muscle damage” (EIMD) and occurs very often when performing exercises with predominantly eccentric actions [1,2].

On the other hand, glutamine is one of the most abundant amino acids in the organism. Glutamine normal plasma levels are 500–750 μmol/L [3]. This amino acid is synthesized by the skeletal muscle and other tissues after transamination from branched chain amino acids. Glutamine plays a structural role as the constituent amino acid in the protein sequence. At the same time, glutamine is the main nitrogen transporter from tissues to excretion in the kidneys [4]. Glutamine plays a key role in acid-base regulation and in gluconeogenesis. Finally, glutamine is the precursor for the biosynthesis of nucleotides. In this context, reduced plasma glutamine levels have been documented in response to different stress situations, such as sustained exercise, trauma and starvation. In these particular stress conditions, increased circulating cortisol stimulates tissular gluconeogenesis from glutamine. It has been documented that in endurance athletes, circulating glutamine decreases seem to occur with a transient immune-depression. Providing glutamine or a glutamine precursor seems to decrease the incidence of sickness in endurance athletes [5]. In this context, the most accepted hypothesis is that skeletal muscle is the main tissue for glutamine biosynthesis. However, muscle activity could alter glutamine availability which is a fuel substrate for immune cell function. Therefore, low plasma glutamine levels as a result of intense exercise are linked to impaired immune function in athletes [6,7]. Nevertheless, in vitro experiments indicate that the reached plasmatic concentrations following exercise do not result in a decreased lymphocyte function. In other words, even in extreme sport events such as a marathon race, the circulating glutamine levels can maintain lymphocyte function.

Although the role of L-glutamine supplementation in the recovery of immune function has been well documented [8,9], the studies regarding restoration of muscle function and reduction of soreness are still scarce [10]. In this context, it has been shown that men supplemented orally with glutamine can attenuate short-term strength loss after an acute bout of eccentric exercise [11]. This result supports the idea that L-glutamine supplementation following intense exercise may favour muscle strength recovery [12,13]. However, the perceived muscle soreness and strength is not evident in certain intervention studies [10]. The presence of other amino acids and the moment of ingestion is a key point to consider. Reduced soreness of the elbow flexors at 24 and 48 h post-exercise was observed in a group of volunteers taking a supplement (3.6 g) containing 12 amino acids (including L-glutamine), ingested pre- and immediately post-eccentric exercise for 4 days [14]. Thus, pre-exercise ingestion of L-glutamine may result in optimal muscle recovery by maintaining physiological concentrations of the amino acid throughout the exercise period.

Finally, glutamine, as well as other amino acids, favours a muscle anabolic state, increasing protein synthesis [15,16]. Street et al. [11] indicated that glutamine supplementation could attenuate the inflammatory response after eccentric exercise. Legault et al. [17] have observed that glutamine supplementation reduces muscle soreness, indicating a possible correlation with less muscle damage. Direct measurement of muscle markers in blood would be required to confirm this hypothesis.

Altogether, there is an existing debate regarding the prevention of post-exercise immune-depression after consuming glutamine supplements. In addition, the possible role for glutamine in stimulating muscle protein synthesis needs further research. The present report is focused on glutamine supplementation in basketball players. Basketball is an intermittent team sport characterized by frequent periods of high-intensity requiring frequent changes of direction, a variety of specific technical skills, and a well-developed jumping ability [18,19]. The capacity to maintain intermittent high-intensity efforts and produce greater strength/power in the legs are generally considered key physical characteristics for high-level basketball players [19]. Therefore, the ability to produce strength, power and speed are important performance characteristics [19]. All of these aspects are developed during specific training sessions, while gym sessions are devoted to resistance training.

Therefore, the available evidence at present is not strong enough to support the use of glutamine supplementation in athletes for immunomodulation and/or anabolic processes. Furthermore, there is even less evidence regarding the role of glutamine in preventing exercise-induced muscle damage. Therefore, the present report aims to test the effect of glutamine supplementation on recovery after eccentric muscle damage in professional basketball players.

## 2. Materials and Methods

### 2.1. Study Design

The study was designed to analyse the effect of glutamine on exercise-induced muscle damage by determining muscle blood markers in a team of professional basketball players (*n* = 12) during a very demanding competition period. The study was a double-blind, placebo-controlled trial. Either glutamine (6 g/day) or placebo was administered orally for 20 consecutive days. The dose and time for glutamine supplementation were selected according to the information provided in references [12,20,21,22]. Seven days before the beginning of the study, a blood sample was obtained (baseline). Then, the crossover study was divided into two consecutive phases, without the knowledge of the players. Participants (*n* = 12) were randomly divided into 2 groups: G (glutamine supplemented, *n* = 6) and P (placebo, *n* = 6). At day 20, groups were crossed (supplementation was changed without the knowledge of the participants) for 20 additional days. The placebo and glutamine capsules were similar in appearance; however, they were identified by a code in the package containing a number plus the letters A or B. Participants and the scientific team ignored the composition of the capsules during the whole intervention period. The glutamine capsules contained lactose and starch plus the corresponding amount of glutamine. The placebo capsules had the same weight as the glutamine capsules, but only contained lactose and starch. This resulted in capsules with similar flavours that made it impossible for them to be distinguished by the participants. Supplementation and placebo were provided by the team doctor early in the morning before the daily training session. On the competition days, capsules were provided at the same hour as in the training days.

The study was designed according to the Declaration of Helsinki for experiments with human beings and was approved by the local Ethics Committee of the University of Basque Country (CEISH/202R/2012). All participants signed a written informed consent form. None of the participants smoked, drank alcohol or were taking medications known to alter the muscular response to exercise. Participants did not take supplements during the intervention. In addition, they displayed a good health status with no clinical or analytical evidence of hepatic, muscular or any other diseases. Finally, participants were free of pathological processes affecting immune system function.

### 2.2. Participants

Participants (professional basketball players) were informed about the research protocol. The study was carried out during the regular season of competition, including matches in Spanish and European leagues. The demographic, physical and physiological characteristics of the players were: 25.3 ± 4.4 years old, 96.8 ± 13.0 kg of body weight, 198.6 ± 9.9 cm of height and 56.5 ± 7.7 ml/kg/min of VO_2_max. These data are expressed as mean ± SD (standard deviation). All subjects completed a medical questionnaire and underwent a cardiopulmonary and electrocardiographic examination, 2 weeks prior to the beginning of the study. A progressive stage exercise test (25 W min^−1^ up to maximal power) was performed on a cycle-ergometer (Monark 818, Bilthoven, Netherlands) to determine maximal oxygen uptake (VO_2_max). Oxygen uptake was measured continuously using a Jaeger ergo-pneumotest (Eos-Compact, Jaeger, Wurzburg, Germany). Heart rate was monitored continuously by electrocardiogram.

All players followed a diet adapted to their particular anthropometric characteristics throughout the whole season, including the study period, and were supervised by the medical group of the team. No supplements, other than glutamine/placebo, were provided during the study. Players followed the same training program of 2 sessions/day: 2 h gym workout in the morning and 3 h of basketball practice in the afternoon. No training was programmed during the 2 weekly days of official matches (Wednesdays and Sundays) played during the 3 weeks of the intervention.

### 2.3. Managing of Blood Samples

Ten hours before blood extraction, a non-fat dinner was planned for participants. After a 30 min rest in supine position, 15 mL of blood samples were obtained at 8:30 a.m. from the antecubital vein of the participants seated in a comfortable position using Vacutainer tubes. WADA guidelines were followed for blood extraction and transportation. Blood was distributed in 2 tubes. Ten mL of blood was placed in one tube containing gel and clot activator to obtain the serum. Additionally, 3–5 mL of blood was extracted and placed in an ethylenediaminetetraacetic acid (EDTA) tube to obtain the plasma. EDTA tubes were inverted 10 times, sealed and stored at 4 °C. The temperature was controlled during transportation by using the specific tag (Libero Ti1, Elpro, Buchs, Switzerland). Plasma was obtained by centrifugation at 2000 rpm for 15 min. Supernatant (plasma) was extracted using a Pasteur pipette, transferred to a sterile tube and stored at −20 °C until the analysis was performed on the 3rd day after blood extraction.

### 2.4. Determination of Muscle Damage Markers

Serum creatine kinase (CK), myoglobin (Mb), aspartate aminotransferase (AST), alanine aminotransferase (ALT), lactate dehydrogenase (LDH), urea, creatinine and total proteins (TP) were determined at baseline (4 days before the beginning of the study), at the beginning (day 1) and end (day 20) of each intervention, according to the crossover design. Testosterone, cortisol and adrenocorticotropic hormone (ACTH) were also analysed following the same extraction schedule. The different parameters were measured by spectrophotometry in an autoanalyser (Hitachi 917, Boehringer Mannheim) as follows: colorimetry (Biuret) for creatinine and TP; enzymatic ultraviolet spectrophotometry for ALT and AST; enzymatic colour test for CK; ultraviolet determination for urea; and immunoturbidimetry for Mb. The enzyme-linked fluorescent assay ELISA performed in a multiparametric analyser (Minividas, Biomerieux, Marcy l’Etoile, France) was used to determine serum cortisol and ACTH levels. ELISA was used to measure serum testosterone (DRG testosterone ELISA kit, DRG Instruments GmbH, Marburg/Lahn, Germany). All determinations were carried out in a certified hospital laboratory with the mandatory technical control.

### 2.5. Statistical Analysis

Statistical analyses were performed using the programming languages Python and R. Data were expressed using 95% intervals for the mean. Normal distribution was checked first by using the Shapiro–Wilk test. Then, a two-way repeated measure analysis of variance (ANOVA) was carried out by the Greenhouse–Geisser test to check the existence of an interaction effect between the supplement and the season training time. Finally, bivariate correlations between circulating enzymes were tested using the Pearson rank order correlation test. A value of *p* < 0.05 was considered as significant.

## 3. Results

Table 1 shows blood muscle markers at the baseline moment and during the intervention, comparing G vs. P. No significant changes were observed, except for AST, which is mainly a hepatic enzyme but is also expressed in the skeletal muscle.

Regarding other haematological parameters, Table 2 shows the data corresponding to the most determining values for physical performance in athletes. Red cell parameters (erythrocyte number, haematocrit and haemoglobin) presented no significant differences in both groups (G and P) compared to the baseline situation. Regarding the white blood cells, including neutrophils and lymphocytes, a significant decrease of the lymphocyte number in the G group was observed compared to the P group and the baseline situation. On the other hand, the neutrophil number increased significantly in the G group compared to the P group and the baseline.

Regarding muscle damage markers such as CK and Mb, Figure 1 shows that CK changed over time in the G group compared to the P group and the baseline situation. A similar pattern was observed for Mb (Figure 2).

Finally, Table 3 shows the levels of hormonal markers corresponding to an anabolic state (testosterone) and a catabolic state (ACTH and cortisol). ACTH (a hormone that stimulates the production of cortisol) decreased significantly in subjects supplemented with glutamine.

## 4. Discussion

Sustained and intense exercise results in muscle damage altering muscle integrity and favouring the release of muscle proteins leading to cell swelling. In addition, the inflammatory response associated with disturbance of the extracellular matrix aggravates muscle function [1,2]. In this context, the most commonly used intracellular proteins that are utilized as muscle damage markers in blood are CK, Mb and LDH.

This study demonstrates that an acute oral supplementation (20 days) with L-glutamine results in a significantly lower release of CK and Mb compared to a placebo supplemented group. LDH displayed a lower release, although the differences with respect to the P group were not significant. Basketball (a sport discipline with predominantly eccentric actions) requires players to perform supple movements and strenuous actions interspersed with active and passive recoveries [23,24]. Altogether, the results indicate that glutamine supplementation seems to exert a positive effect in the recovery after eccentric exercises, reflected in lower circulating levels of CK and Mb, during a period of very demanding training and competition.

In this context, Souglis et al. [25] compared the inflammatory responses and muscle damage in male top players from four different disciplines: soccer, basketball, volleyball and handball. Muscle damage markers (represented by serum increases in CK and LDH activities) revealed that basketball resulted in an intermediate level of stress. Results from the present report and from other laboratories [8,26] corroborate these findings. In addition, sport disciplines that are predominantly eccentric require a full and rapid recovery after competition. Therefore, interventions that help to diminish the effects of muscle damage are helpful for players, helping them to maintain an adequate level of performance and an optimal adaptation to the subsequent training sessions [1,2].

Parry-Billings et al. [27] studied the effect of glutamine supplementation in five adults of 22–41 years old. Results from this report indicated that periods of very heavy training are associated with a chronic reduction in plasma concentrations of glutamine. It has been suggested that this decrease may result in a low immune response observed in many endurance athletes. In addition, it is known that muscular glutamine levels are related to the rate of protein synthesis [28,29]. In this vein, low plasmatic concentrations of glutamine have been documented in different clinical situations, including major surgery, burns, starvation and sepsis [27,30,31,32]. Therefore, it can be hypothesised that requirements for glutamine need to be increased in these conditions in order to modulate the activity of the immune system and stimulate cellular divisions involved in muscle repair processes. In this context, glutamine is the main fuel of the immune cells and for repair processes.

Regarding leukocytes, the number and percentage of neutrophils and lymphocytes were more or less stable, with modest variations during the study in the G and P groups. This may suggest that an adequate training and recovery process was carried out, allowing for leukocyte stability [33].

Regarding anabolic/catabolic hormones, a non-significant decrease in cortisol was observed with glutamine supplementation. This observation suggests that the recovery effect of glutamine on muscle damage might be not due to direct neuroendocrine effects. In this context, a relevant result was that ACTH (the main stimulating hormone for cortisol release) was lower in the G group than in the P group. Basal ACTH levels are used as indicators of acute stress [34]. A previous study in a similar group of basketball players [35] indicated dramatic increases of ACTH during intense sustained exercise, favouring cortisol release. In the present report, we have observed sustained levels of basal ACTH in the P group and a decrease in the G group.

On the other hand, high serum cortisol levels indicate accumulated stress as a result of intense and/or sustained exercise [34,35,36]. In other words, high cortisol levels appear after high-intensity actions and long duration exercises. In addition, low cortisol levels seem to be necessary to control the post-exercise inflammatory response [37] that is instrumental for optimal muscle repair/regeneration during the recovery process, allowing for improvement in performance [38]. In this study, both groups G and P maintained serum levels of cortisol. This observation could be in accordance with the fact commented on previously that the players have followed an adequate training and recovery process.

A similar result was observed for testosterone levels, confirming the optimal effect of the supplementation. The pattern of testosterone in this study reflects a good recovery, maintaining an anabolic profile which is instrumental during demanding periods of training and competition.

In summary, the study presents some limitations. For instance, we did not take into account the court position of the players due to the low number of participants. In addition, cytokine determination could help to monitor more accurately the status of the immune system. We plan to address these and other questions in future research. In conclusion, the data presented show that glutamine supplementation results in a decrease of circulating muscle damage markers accompanied by an adequate balance between the response of the catabolic and anabolic hormones and the stability of leucocyte cell numbers. We hypothesize that the control of these specific parameters could help to prevent the inflammation and stress provoked by highly strenuous exercise. From a practical point of view, glutamine supplementation could help in recovery after intense and demanding eccentric exercises that produce muscle damage with a high risk of lesions.

## Figures and Tables

**Figure 1 nutrients-13-02073-f001:**
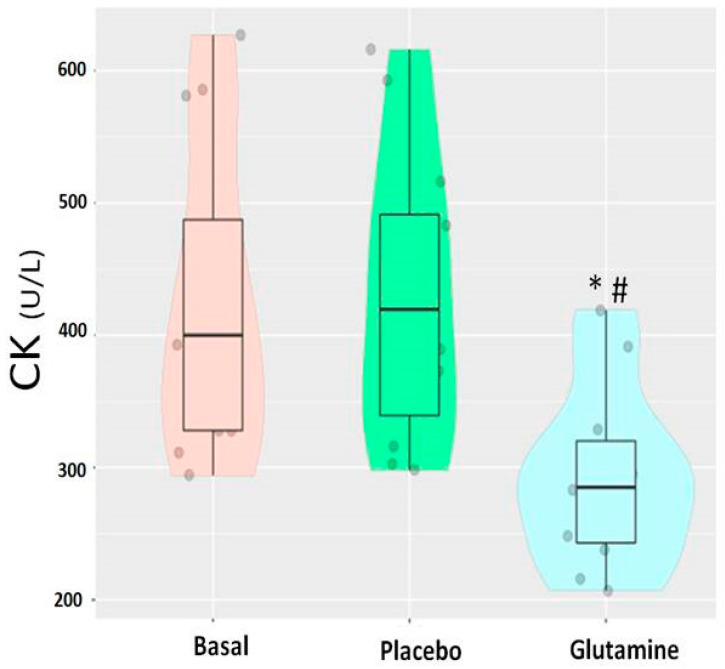
Creatine kinase levels (CK) in the baseline situation (basal), placebo group and glutamine supplemented group. (*) Significant values (*p* < 0.05) comparing glutamine vs. basal. (*^#^*) Significant values (*p* < 0.05) comparing glutamine vs. placebo. The clear similarity between baseline levels and placebo indicates that there was no placebo effect (a t-test showed no significant differences between these two groups, *p* = 0.9).

**Figure 2 nutrients-13-02073-f002:**
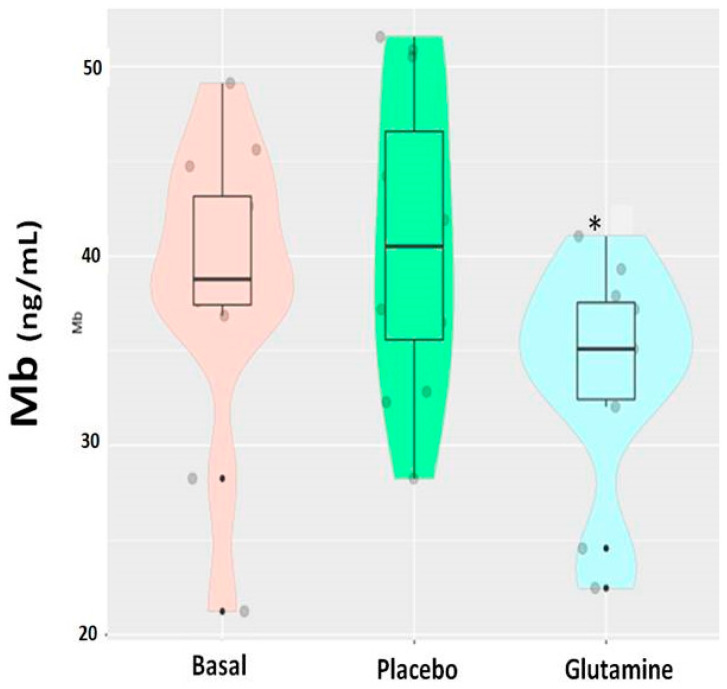
Myoglobin levels (CK) in the baseline situation (basal), placebo group and glutamine supplemented group. (*) Significant values (*p* < 0.05) comparing glutamine vs. basal and comparing glutamine vs. placebo. The clear similarity between baseline levels and placebo indicates that there was no placebo effect (a t-test showed no significant differences between these two groups, *p* = 0.48).

**Table 1 nutrients-13-02073-t001:** Blood muscle damage markers.

Parameter	B	G	P
LDH (U/L)	374.2 ± 29.81	384.4 ± 47.70	396.8 ± 36.90
AST (U/L)	33.38 ± 4.43	30.13 ± 5.08 *	33.08 ± 4.4
ALT (U/L)	24.8 ± 3.41	23.9 ± 2.72	25.42 ± 3.62
Urea (mg/dL)	43.8 ± 6.04	42.9 ± 5.98	42.58 ± 5.40
Creatinine (mg/dL)	1.25 ± 0.05	1.23 ± 0.04	1.29 ± 0.07
TP (g/L)	6.98 ± 0.56	7.08 ± 0.45	6.97 ± 0.49

B: baseline, G: glutamine (6 g/day for 20 days), P: placebo (20 days). Significant differences: * *p* < 0.05 comparing G vs. B. Abbreviations used: ALT (alanine aminotransferase), AST (aspartate aminotransferase), LDH (lactate dehydrogenase), TP (total proteins).

**Table 2 nutrients-13-02073-t002:** Results referring to circulating cell number and related parameters.

Parameter	B	G	P
Erythrocytes (10^6^/uL)	4.88 ± 4.32	4.86 ± 3.98	4.95 ± 4.51
Hematocrit (%)	44.56 ± 1.3	44.52 ± 1.9	44.75 ± 2.2
Hemoglobin (g/dL)	14.63 ± 0.56	14.54 ± 0.78	14.32 ± 0.62
WBC (10^3^/uL)	6.13 ± 1.15	5.81 ± 0.98	6.73 ± 1.11
Neutrophils (%)	47.98 ± 7.2	59.31 ± 8.1	44.18 ± 4.5
Neutrophils (10^3^/uL)	2.94 ± 0.25	3.02 ± 0.19 ***	3.64 ± 0.19
Lymphocytes (%)	46.76 ± 6.8	37.42 ± 5.1 ***^,*#*^	53.75 ± 5.7
Lymphocytes(10^3^/uL)	2.87 ± 0.18	2.17 ± 0.15	2.94 ± 0.20

B: baseline, G: glutamine (6 g/day for 20 days), P: placebo (20 days). Significant differences: * *p* < 0.05 comparing G vs. B, *^#^ p* < 0.05 comparing and G vs. P. Abbreviations used: WBC (white blood cells).

**Table 3 nutrients-13-02073-t003:** Hormonal markers analysed in the different groups.

Hormone	B	G	P
Cortisol (ug/dL)	17.67 ± 2.5	17.21 ± 3.2	18.93 ± 3.4
Testosterone (ng/dL)	5.87 ± 1.11	5.98 ± 1.04	6.07 ± 0.98
ACTH (pg/mL)	78.7 ± 22.49	53.7 ± 22.90 *^,#^	76.5 ± 23.02

B: baseline, G: glutamine (6 g/day for 20 days), P: placebo (20 days). Significant differences: * *p* < 0.05 comparing G vs. B and ^#^ *p* < 0.05 comparing G vs. P. Abbreviations used: ACTH (adrenocorticotropic hormone).

## Data Availability

The data that support the findings of this study are available from the corresponding author, upon reasonable request.

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
