# Peer review of "Effect of Glutamine Supplementation on Muscular Damage Biomarkers in Professional Basketball Players"

_nutrients, 2021, doi:10.3390/nu13062073_

Round 1
Reviewer 1 Report
The authors conducted a crossover study design in population of basketball players to reveal the effect of uL-glutamine supplementation on systemic immune function and muscle damage. The experiments were carefully prepared and the measurement of selected parapeters was done in an appropriate manner. The results are reliable but some things are missing.
Introduction:
Overall, there is a presented rationale for the study with regards to L-glutamine supplementation, but it would be helpful if the rationale for the effect in baskteball training and competition were made clearer.
Is there any diffrences in specifity of training in basketball depending on court possition on this stage of competition that coud indicte specific tissue response - C, SF, PF, SG etc...???
Is there any merit in mentioning the effect of low L-glutamine status on physical performance at this point due to the nature of the investigation?
Methods
"Participants were supplemented with 6 g/day of glutamine (G group) or placebo (P group) for 40 days in 23 a crossover study design (20 days with glutamine + 20 days with placebo and vice versa)"How this amount glutamine was selected andy why supplementation protocol lasted for 20 days, not longer...???
Why only creatine kinase (CK), myoglobin (Mb), aspartate aminotransferase (AST), alanine aminotransferase (ALT), lactate dehydrogenase (LDH), urea, creatinine and total proteins (TP) were determined? What about musscle specific (myocinse) or inflamation specific (cytocines) proteins???
Results
Figure 2,4 dont looks clear. Could the authors consider presenting figure 2, 4 in much in a much more accessible form???
Line 148 How long plasma wos stored until analysis???
Discussion
The discussion provides a nice overview of the results, but there needs to be some development of the mechanistic underpinnings and practical implications. Furthermore limitations of the studie are missin but from the text we can conclude that there are some.
I hope that the authors find the above comments helpful and in the constructive manner they are intended.
Author Response
Estimado Revisor
Agradecemos la cuidadosa revisión y las constructivas sugerencias. Hemos procurado atender a sus propuestas por lo que creemos que el manuscrito ha mejorado sustancialmente tras realizar los cambios sugeridos.
Best regards

Reviewer 2 Report
The authors performed an interesting study. I do feel the manuscript needs some improvements to make sure the study and the results are well described.
Major comments
When reading the introduction you assume muscle soreness (thus subjectively measured in the participants) will also be measured. Unfortunately I do not see these results in the manuscript. Biomarkers for muscle damage in blood are not always leading to direct or noticeable benefits by the participants themself on their muscle soreness, and therefore motivation/compliance of using supplementation like glutamine around exercise will be lower. If you have data of muscle soreness and or fatigue levels, I would strongly recommend to add this to the manuscript.
Line 103-104: Supplementation was changed without knowledge of participants. Were the glutamine and placebo that much alike then? I would like to read more information on how that was done. What was the placebo exactly? How much gram was that? How was it made similar to the glutamine supplementation for smell, taste, consistency etc.? When you switch halfway through the study it seems to me almost impossible that no participant had suspicions about that their supplementation was changed. And where there adverse events reported?
It is also not clear when the participants received the supplementation. You describe in your introduction that it is good to take before exercise, so I assume it was given then, but it is not mentioned in the methods section.
Please include an analysis whether there was a sequence effect. And what value was considered as baseline in the group that first received placebo? The baseline measurement at the beginning or after 20 days of using placebo?
In the text around tables and figures it seems that you did not check whether a placebo effect occurred between baseline and after 20 days of using placebo. Is that correct? If so, could you add this? Just to be sure.
In the methods section you mention that oxygen uptake and heart rate was measured continuously. Could you present those (average) findings at well?
Please give some more details about the participants in the other studies that are mentioned in the discussion (e.g. age, amateur or elite level etc.)
Is your finding of the leukocytes similar to findings from previous studies?
Minor comments
In line 196 you use the phrase ‘’after 20 days of supplementation’’. Please make sure you use that ‘’changes over time’’ in the whole results section. That makes it much clearer.
Table 3. delete one dot ‘’.’’ in line 219.
Author Response

(The authors gave the same response as above.)

Round 2
Reviewer 2 Report
In the response you say that the supplements were identified by letters A and B. That seems to be essential information to be in the paper as well. I feel this sentence ‘’Participants and the scientific team ignored the composition of the capsules during the whole intervention period’’ is not very convincing regarding trustworthiness.
Please perform a statistical analysis to confirm that indeed there was no sequence effect by comparing the changes in the one group to the other one.
Thank you for adding the sentence about the possible placebo effect. Please include the P-values as well. This can also be beneficial for future systematic reviews and/or meta-analyses that might want to include your study.
So where these basketball players mentioned in the article of Córdova-Martínez et al, 2010 also playing at high competition levels? How many were included in that study? Please mention shortly in section 2.2. participants that the participants from your study are professional athletes.
Please mention in line 260-263 that the leukocyte levels were similar to those found in other study and add references to that section.
Line 106-107: according to what?
Line 112-118: The spelling of the new addition really needs extensive editing of English language and style. Also, were similar amounts of lactose and starch then used in the placebo and the glutamine supplementation?
Author Response
Dear Reviewer,
We thank you for the second review of our manuscript entitled "EFFECT OF GLUTAMINE SUPPLEMENTATION ON BIOMARKERS OF MUSCLE DAMAGE IN PROFESSIONAL BASKETBALL PLAYERS".
We added the responses to the new comments to it. The changes have been highlighted in blue in the manuscript.